# On the Development of Nanocomposite Covalent Associative Networks Based on Polycaprolactone and Reduced Graphite Oxide

**DOI:** 10.3390/nano12213744

**Published:** 2022-10-25

**Authors:** Alberto Vallin, Daniele Battegazzore, Giacomo Damonte, Alberto Fina, Orietta Monticelli

**Affiliations:** 1Dipartimento di Chimica e Chimica Industriale, Università degli Studi di Genova, Via Dodecaneso 31, 16146 Genoa, Italy; 2Dipartimento di Scienza Applicata e Tecnologia, Politecnico di Torino-Sede di Alessandria, Viale Teresa Michel, 5, 15121 Alessandria, Italy

**Keywords:** PCL, biopolymers, vitrimers, rGO, nanocomposites

## Abstract

In this work, the development of nanocomposite systems based on reduced graphite oxide (rGO) was combined with the development of crosslinked materials characterized by dynamic covalent bonds, i.e., a covalent associative network, starting from *ad-hoc* synthesized hydroxyl terminated polycaprolactone (PCL-OH). The crosslinking reaction was carried out using methylenediphenyl diisocyanate (MDI) to create systems capable of bond exchanges via transesterification and transcarbamoylation reactions, in the presence of stannous octoate as a catalyst. The above materials were prepared at two different temperatures (120 and 200 °C) and two PCL-OH:MDI ratios. FT-IR measurements proved the formation of urethane bonds in all the prepared samples. Crosslinking was demonstrated by contacting the samples with a solvent capable of dissolving the star-shaped PCL. These tests showed a significant increase in the crosslinked fraction with increasing the temperature and the PCL-OH:MDI ratio. In order to evidence the effect of crosslinking on rGO dispersion and the final properties of the material, a nanocomposite sample was also prepared using a linear commercial PCL, with the nanofiller mixed under the same conditions used to develop the crosslinked systems. The dispersion of rGO, which was investigated using FE-SEM measurements, was similar in the different systems prepared, indicating that the crosslinking process had a minor effect on the dispersibility of the nanofiller. As far as the thermal properties are concerned, the DSC measurements of the prepared samples showed that the crosslinking leads to a decrease in the crystallinity of the polymer, a phenomenon which was particularly evident in the sample prepared at 200 °C with a PCL-OH: MDI ratio of 1:1.33 and was related to the decrease in the polymer chain mobility. Moreover, rGO was found to act as a nucleating agent and increase the crystallization temperature of the nanocomposite sample based on linear commercial PCL, while the contribution of rGO in the crosslinked nanocomposite samples was minor. Rheological measurements confirmed the crosslinking of the PCL-OH system which generates a solid-like behavior depending on the PCL-OH:MDI ratio used. The presence of rGO during crosslinking generated a further huge increase in the viscosity of the melt with a remarkable solid-like behavior, confirming a strong interaction between rGO and crosslinked PCL. Finally, the prepared nanocomposites exhibited self-healing and recyclability properties, thus meeting the requirements for sustainable materials.

## 1. Introduction

PCL represents one of the most promising and widely used bioplastics and improving its properties is an extremely significant and studied aspect [1,2]. There are two strategies that can be applied to achieve this objective: modification of the polymer structure, such as the geometry as well as the end functionalities [3,4,5], and blending with other polymers [6] or with suitable fillers/nanofillers [7], such as graphene-related materials (GRM) [4,8,9,10]. As the preparation of composite/nanocomposite systems is concerned, the main issue in the development of the above-mentioned materials is related to the dispersion of the filler in the polymer matrix. As such, different strategies were explored to improve the affinity between the polymer and the surface of GRM, and thus their dispersibility in the polymer matrix. In particular, partially oxidized nanosheets such as graphite/graphene oxide (GO) were applied to promote interactions with polar macromolecules, or, alternatively, the polymer was modified to promote non-covalent bonding onto graphitic surfaces [8,9,10,11]. When using GO, oxidized groups on the surface may be conveniently exploited for covalent polymer grafting, in particular by the in situ polymerization method, which enabled the preparation of nanocomposites based on several polymers, including PCL [8]. Thermally reduced graphene oxide (TRGO) was also used to develop nanocomposites based on PCL. Indeed, the reduction promotes partial recovery of the disruption of graphene sp^2^ structure, leading to better electrical and thermal conductivity, which are desirable properties in several electric and electronic applications. The conventional solution casting method was reported for the preparation of PCL/TRGO systems [12]. Recently, an effective approach to promote interactions with non-functionalized GRM was also reported, based on the modification of the chemical structure of the polymer by inserting appropriate functionalities, such as pyrene groups [4,13]. Concerning the properties of the systems based on GRM and PCL, various studies evidenced that the nanofiller acted as a nucleating agent by increasing the polymer crystallization temperature [4,12,13,14]. In addition, GRM dispersion in PCL was found to significantly affect the mechanical properties, increasing the storage modulus [10]. However, some intrinsic properties of the material, such as functionality and solvent resistance, are not affected by the addition of the nanofiller to the polymer matrix. To enhance solvent resistance and mechanical performance, especially at high temperatures, dynamic crosslinking of polymers has been recently developed either based on dissociative or associative networks [15,16].

Dissociative networks are based on cleavable/reformable bonds, often thermally activated such as in Diels-Alders (DA) adducts, widely exploited in several polymers, including PCL to prepare stimulus-response systems that are generally applied in the biomedical sector [17,18]. The preparation of PCL-based DA networks foams was investigated by Houbben et al. [18] starting from a mixture of low molar mass PCL stars bearing furan or maleimide as end-groups, reacting to produce the crosslinked network of PCL throughout the foaming process, which was accomplished by a solvent-free supercritical CO_2_ foaming process. The same reaction was used to synthesize PCL-poly(ethylene oxide) (PEO) dynamic networks, starting from 4-arm star-shaped PEO and PCL end-capped with furan and maleimide moieties, respectively [19]. Indeed, thanks to the introduction of thermo-reversible Diels–Alder adducts within the covalent network, the above material was found to be easily recycled while preserving the shape-memory performances. More recently, Cao et al. [20] prepared composite PCL-based systems by crosslinking through DA bonds, and then the resultant network was also connected to maleimide functionalized Al_2_O_3_ particles. The resultant covalent adaptable composite network was de-crosslinked at the retro-DA reaction temperature, leading to a great decrease of viscosity in favor of compounding and molding.

Covalent associative networks (CAN) are based on exchangeable bonds, allowing changes in the topology of the networks to occur without decreasing the crosslinking density. These systems were first reported by Leibler et al. [21] and referred to as vitrimers, based on the analogies with the rheological behavior of inorganic glasses at high temperatures. Examples of bond exchange mechanisms include transesterification, transcarbamoylation, transamination, etc. [22,23]. Despite the increasing interest in bio-based polymers and the promising properties that can be obtained by applying the described approach, very few works have applied dynamic covalent crosslinking to such materials. Hillmyer et al. [24] reported on the preparation of vitrimer based on hydroxyl-terminated star-shaped polylactide, crosslinked by diisocyanate, which showed healing properties by compression molding via catalyzed transesterification reactions. PCL was also turned into a covalent network by the reaction of linear hydroxyl-terminated PCL and trifunctional isocyanate, in the presence of a catalyst to promote transesterification reactions [25].

Changing the conformation of the polymer, i.e., going from linear to branched or star-shaped macromolecules, can significantly increase the functionality of PCL and allow precise tailoring of the CAN properties. Besides the design and development of vitrimer networks, it is worth underlying that combining dynamic crosslinking of the polymer with the inclusion of nanoparticles could further improve the final performance of the material. On this basis, nanostructured CAN based on PCL and reduced graphite oxide (rGO) has been developed for the first time in the present work. In particular, the work is based on the previous synthesis of a star-shaped PCL of low molecular weight terminated with -OH groups. The crosslinking reaction was carried out with a very simple and fast reaction, that involved the application of MDI with two different PCL-OH:MDI ratios and two different temperatures (120 °C and 200 °C). The nanocomposite systems were prepared under the same conditions and with the addition of 1 wt.-% rGO to the reaction mixture (Figure 1). To demonstrate the effect of crosslinking, a sample was also prepared using a commercial linear PCL. The prepared samples were characterized by FT-IR, DSC, TGA, and rheological measurements. In addition, gel fraction, swelling properties as well as recycling and healing were evaluated by comparing the behavior of the neat crosslinked samples with those based on rGO and the system prepared from a commercial linear PCL.

## 2. Materials and Methods

### 2.1. Materials

ε-Caprolactone, pentaerythritol, tin octanoate (Sn(Oct)_2_), toluene (anhydrous, purity 99.7%), dichloromethane (DCM), chloroform, methanol and methylene diphenyl diisocyanate (MDI) were purchased from Sigma Aldrich^®^ (Milan, Italy). Prior to use, ε-caprolactone was purified by vacuum distillation over CaH_2_. All other reagents were of analytical grade and were used without purification. Reduced graphite oxide (rGO) with a surface area of 196 m^2^/g (BET), an oxygen content of 2.0% (XPS), and a Raman value of I_D_/I_G_ 1.35 at 514 nm, were provided by Avanzare Innovación Tecnólogica (Navarrete (La Rioja), Spain). The preparation method [26] and full characterization [27] was previously reported elsewhere. Commercial grade PCL CAPA^®^ 6500 (Mw = 50,000 g/mol) was purchased from Perstorp (Malmö, Sweden).

### 2.2. Synthesis of PCL-OH

The synthesis of star-shaped hydroxyl-terminated PCL (hereafter referred to as PCL-OH) was carried out by ring-opening polymerization (ROP) of ε-caprolactone in bulk using Sn(Oct)_2_ as catalyst according to a previously described procedure [4,13,28]. The initiator, namely penthaerythritol, was adjusted to give arms with a molecular weight of 2000 g·mol^−1^. Specifically, 8.3 g of the monomer was added to a 50-mL round-bottom flask with two-necks, followed by 142 mg of the initiator. To facilitate complete solubilization of the initiator, the system was placed in an oil bath and heated at 80 °C with stirring. Then, the temperature was increased to 120 °C and the reaction was started by adding 60 μL solution of Sn(Oct)_2_ in anhydrous toluene (100 mg/mL) under an argon atmosphere. The amount of catalyst was chosen so that the ratio of [ε-CL]/[Sn(Oct)_2_] was 5000. The reaction was performed at 120 °C for 24 h. The mixture was then cooled to room temperature and the polymer obtained was dissolved in 2 mL of DCM. The product was then precipitated in 200 mL of cold methanol with vigorous stirring, filtered through a Büchner funnel, washed several times with cold methanol, and dried at 40 °C for 72 h.

^1^H-NMR chemical shifts of PCL-OH (Appendix A): 4.09 ppm (-CH_2_- pentaerythritol, s); 4.05 ppm (-CH_2_-, t); 3.65 ppm (-CH_2_-OH PCL chain terminal, t); 2.29 ppm (-CH_2_-, t); 1.64 (-CH_2_-, m); 1.38 (ppm -CH_2_-, m).

IR signals of PCL-OH (Appendix A): 2945 cm^−1^ (asymmetric -CH^2^- stretching); 2870 cm^−1^ (symmetric -CH_2_- stretching); 1725 cm^−1^ (symmetric >C=O stretching); 1297 cm^−1^ (-C-O- and -C-C- stretching); 1242 cm^−1^ (asymmetric -C-O-C- stretching); 1179 cm^−1^ (symmetric -C-O-C- stretching).

### 2.3. Preparation of the Crosslinked Systems

For the synthesis of crosslinked systems, a reaction between the star hydroxyl terminated PCL and MDI was carried out. Four different samples were prepared at different temperatures (120 °C and 200 °C) and two molar ratios of PCL-OH:MDI, namely 1:1 and 1:1.33 (Table 1), i.e., under conditions that allow the production of a material with a suitable concentration of -OH groups to promote bond exchange.

Specifically, 3 g of PCL-OH was placed in a tubular reactor (2 cm inner diameter and 23 cm length) under an argon atmosphere and placed in an oil bath heated to 120 °C or 200 °C. After the complete melting of the polymer, 21.3 μL of catalyst solution (Sn(Oct)_2_ in toluene anhydrous 100 mg/mL) and MDI (92 mg for 1:1 and 123 mg for 1:1.33) were added. The mixture was prepared using a laboratory internal mixer equipped with a mechanical stirrer, type RZR1 (Heidolph Instruments GmbH & Co, Schwabach, Germany). Both the catalyst and MDI were added under an argon atmosphere. The reactions started instantly with the additions of MDI and were performed for 30 min. After cooling at room temperature, the samples were dried at 30 °C under vacuum for 72 h.

### 2.4. Preparation of Nanocomposites

The same conditions and equipment as described previously were applied to prepare the nanocomposites (Table 1). In this case, 1 wt.-% of rGO with respect to the polymer, was added and dispersed in the molten polymer before the catalyst and MDI were added. A nanocomposite sample was prepared starting from the commercial linear PCL CAPA^®^ 6500 (hereafter referred to as PCL) by adding 1 wt.-% of rGO to the polymer at 200 °C and using the same equipment as for the preparation of the crosslinked sample.

The samples were defined by specifying the molar ratio of PCL-OH to MDI (1:1 or 1:1.33) and by indicating the crosslinking temperature (120 °C and 200 °C) in the code. The example PCL-OH_MDI_1:1_200 indicates a sample prepared with a molar ratio PCL-OH:MDI of 1:1 and prepared at a temperature of 200 °C. In the case of nanocomposite systems, a “G” was added to the code. For the system prepared from PCL, the code was PCL-G.

### 2.5. Characterization

Differential scanning calorimetric (DSC) measurements were carried out under a continuous nitrogen purge on a Mettler calorimetric apparatus, model DSC1 STAR^e^ in the temperature range from −100 to 150 °C. A scanning rate of 10 °C/min was used both on heating and cooling. Both calibrations of heat flow and temperature were based on a run in which one standard sample (indium) was heated through its melting point. The crystallinity (*X_c_*) of PCL follows Equation (1):(1)$$X_{c}\left(\%\right)=\frac{\Delta H_{m}}{\Delta H_{m}^{0}}\times 100\%$$
where Δ*H_m_* is the measured heat of fusion and $\Delta H_{m}^{0}$ is the melting enthalpy of the 100% crystalline PCL (139 J/g) [29].

Thermal gravimetrical analysis (TGA) was performed by using a STAR^e^ System Mettler thermobalance under a flow of nitrogen of 80 mL/min. The weight loss of the samples (having initial masses of ca. 10 mg) was measured from room temperature to 800 °C at a heating rate of 10 °C/min.

FT-IR analysis was performed with a Bruker “Vertex 70^®^” operating in ATR mode in the range 400–4000 cm^−1^.

A Zeiss Supra 40 VP field emission scanning electron microscope (FE-SEM) equipped with a backscattered electron detector was used to examine the nanocomposite morphologies. The specimens were submerged in liquid nitrogen for 30 min and fractured cryogenically. All samples were thinly sputter-coated with carbon using a Polaron E5100 sputter coater.

The gel fraction (*GF*) and swelling ratio (*SR*) were analyzed using DCM, a solvent capable of dissolving the star-shaped polymer. For this purpose, about 70 mg of the samples were accurately weighed and contacted with 3 mL of DCM for 24 h. The samples were then removed and first dried at 25 °C and atmospheric pressure for 6 h and then stored in an oven at 30 °C under vacuum for 24 h. The samples were weighed, and the *GF* was calculated using Equation (2).
(2)$$GF=\frac{M_{d}}{M_{i}}\cdot 100$$
where *M_d_* is the weight of the dried sample and *M_i_* is its initial weight.

To calculate the *SR*, samples were weighed after 24-h contact with DCM, and Equation (2) was applied.
(3)$$SR=\frac{M_{wet}-M_{d}\;}{M_{d}}\cdot 100$$
where *M_wet_* is the weight of the swollen polymer and *M_d_* is the weight of the dried sample.

Recycling tests were performed on samples with a diameter of 1 cm and a thickness of 1 mm. The specimens were divided into fragments and placed in a brass mold of 3 × 3.5 cm and 250 μm and then pressed for 30 min at 6000 kgf at 200 °C.

Similarly, self-healing tests were carried out using a hot press. In particular, a sample with a size of 3 × 3.5 cm and a thickness of 250 μm was cut with a knife, then placed in a mold and cured by compression molding for 30 min at 200 °C and 6000 kgf.

The rheological properties of the materials were analyzed using an ARES rheometer fitted with a 25 mm parallel plate geometry. The sample for the characterization was obtained from a hot compression process in a 1 mm thick mold. The materials were previously dried in a vacuum oven overnight at 40 °C and then pressed at the same temperature at which they were crosslinked (120 °C or 200 °C), 50 bar hydraulic pressure for 10 min. The disks were kept dry until the rheological tests.

Dynamic strain sweep tests were firstly carried out to confirm the linearity of the viscoelastic region up to 1% strain. Subsequently, frequency sweeps were carried out to determine the complex viscosity (η*) and storage modulus (G’) over a frequency range of 0.1–100 rad/s from 80 °C to 200 °C at intervals of 30 °C. Rheological tests were performed with nitrogen flux to avoid hydrolytic degradation.

## 3. Results

### 3.1. FT-IR, Swelling, Thermal and Rheological Analysis of Neat Polymers and Dynamic Networks

To investigate the chemical structure of the developed polymers, an FT-IR characterization was performed comparing the spectra of the neat PCL-OH (Figure 2A) with those of the crosslinked systems prepared at two temperatures and two different PCL-OH/MDI ratios.

All samples showed the typical bands of PCL with peaks at 2950 cm^−1^ and 2870 cm^−1^ (symmetrical/unsymmetrical stretching Csp^3^-H bonds in methylenic unit), 1728 cm^−1^ (carbonyl stretching), 1296 cm^−1^ (C-O and C-C stretching), 1241 cm^−1^ and 1170 cm^−1^ (unsymmetrical/symmetrical C-O-C stretching) [30]. In addition, in the neat star-shaped polymer, a broadband is clearly visible in the range between 3600 cm^−1^ and 3200 cm^−1^, which can be assigned to the stretching of the end -OH groups. Indeed, in all the crosslinked samples, the intensity of the above peak was much lower, indicating a lower concentration of hydroxyl groups. Moreover, one new signal appears at 1535 cm^−1^ in these samples (Figure 2B), which can be attributed to the N-H bending of the secondary amide groups [31]. Based on the above results, the occurrence of the reaction between the hydroxyl groups of the star-shaped PCL and MDI is confirmed (Figure 2C).

It is worth underlining that, as described in the literature for other polyesters, such as polylactic acid [23,24], i.e., polymers with a similar chemical structure to PCL, the presence of the catalyst can activate transesterification and transcarbamoylation reactions involving free hydroxyl functionalities of the system (Figure 2D and Appendix A), thus forming a covalent associative network.

To evaluate the formation of a network as a result of the reaction between the hydroxyl end groups of the star-shaped polymer and MDI, the prepared samples were contacted with a solvent, i.e., DCM, capable of dissolving the neat unreacted PCL. The gel fraction (*GF*) of the systems prepared at different temperatures and PCL-OH:MDI ratios are shown in Table 2.

Indeed, by comparing the behavior of the different systems, the influence of the preparation conditions on the cross-linking of the material was verified. Using the same solvent and starting from the washed system, i.e., the polymer without the non-crosslinked portion, the swelling ratio (*SR*) was also evaluated. Considering the data reported in Table 2, it is evident that the crosslinked fraction increased with increasing the preparation temperature with the MDI and the PCL-OH:MDI ratio, it being 24% and 73% for PCL-OH_MDI_1:1_120 and PCL-OH_MDI_1:1.33_200, respectively. Thus, this result indicates that it is possible to tune the final properties of the materials by changing the conditions of preparation. The swelling ratio (*SR*) does not seem to follow the same trend as reported for *GF*, since the values for all samples were around 3000%. It is possible to hypothesize that increasing the temperature and the PCL-OH:MDI ratio increased the crosslink conversion without significantly changing the network structure.

To further investigate the crosslinking of the different formulations, rheological measurements were carried out as a function of the angular frequency (ω) and temperatures between 80 and 200 °C, after compression molding was successfully validated for all prepared formulations. The typical behavior of a molten polymer displays a constant decrease of the storage modulus (G’) as shear frequency decreases. Dynamic networks based on PCL-OH exhibited different properties depending on the preparation temperature and the PCL-OH:MDI ratio (Figure 3).

In PCL-OH_MDI_1:1_120, G’ is indeed decreasing with approx. constant slope with decreasing frequency, which is coherent with a limited crosslinking density, in agreement with gel fraction and swelling ratio. Signs of slightly higher crosslinking densities are observed on G’ plots for both PCL-OH_MDI_1:1.33_120 and PCL-OH_MDI_1:1_200 plots, showing decreasing slope in G’ (Figure 3c) plot at low frequency and higher values for complex viscosity (η*, Figure 3b,c). In PCL-OH_MDI_1:1.33_200, G’ values at low frequency are almost constant, reflecting a solid-like behavior, coupled with significantly higher complex viscosity compared to PCL-OH_MDI_1:1, thus confirming the highest crosslinking density in the melt among the four formulations, in agreement with gel fraction analyses. It is also worth mentioning that the decrease of G’ and η* plots with increasing testing temperature is remarkable in materials prepared at 120 °C and significantly lower in materials prepared at 200 °C. This confirms the preparation at higher temperatures allows better stability of the network, likely explained by better homogeneity and higher crosslinking density.

The DSC traces of neat PCL-OH and the dynamically crosslinked materials prepared from this polymer are shown in Appendix A while the thermal data are summarized in Table 3, where data related to PCL are also given. The neat PCL showed typical behavior of a semi-crystalline polymer, with crystallinity (*X_c_*), melting (T_m_), and crystallization temperatures (T_c_) in agreement with the data reported in the literature for a polymer with similar characteristics [32]. It is worth underlining that *X_c_* of the star-shaped polymer is higher than that of the linear PCL. Indeed, although a decrease in crystallinity is expected due to the star geometry [32], the low molecular weight of PCL-OH (about 2000 g/mol for each arm) promotes polymer structuring, compensating for the decrease due to the branched structure.

A decrease in T_c_ and *X_c_* was observed in the case of the crosslinked samples, which is particularly evident for PCL-OH_MDI_1:1.33_200, where T_c_ decreased from 28 to 17 and *X_c_* from 58% to 46% for PCL-OH and the crosslinked sample, respectively. In order to explain the above results, it is necessary to take into account that the studied systems consist of a crosslinked part and free polymer chains characterized by -OH end groups which can participate in the transesterification reaction during heating (Figure 2). Thus, as previously reported, the highest degree of crosslinking in the sample prepared at the highest temperature and with the highest PCL-OH:MDI ratio led to the greatest decrease in crystallinity, as the chain mobility of most of the sample is reduced.

Based on the above results, the development of composite systems focused on samples prepared at 200 °C, as this temperature was found to ensure the most efficient crosslinking.

### 3.2. Morphological, Rheological and Thermal Analysis of Nanostructured Dynamic Networks

Morphological characterization, carried out by FE-SEM measurements, was performed to verify the dispersion of rGO in the polymer matrix, comparing the crosslinked nanocomposite samples (PCL-OH_MDI_1:1_200 and PCL-OH_MDI_1:1.33_200) with that based on neat PCL (PCL-G).

The micrographs reported in Figure 4 show the fracture surface of the prepared samples. All systems examined were characterized by a homogeneous distribution of rGO layers without any visible agglomerates. As particularly evident in the high-magnification micrograph (Figure 4e), there was a significant adhesion of the nanofiller to the polymetric matrix and the absence of voids, demonstrating the good compatibility of rGO with the polymer. It is worth underlining that this morphology is similar for all the samples studied, both crosslinked and that based on the neat PCL (Figure 4c), proving that crosslinking did not significantly affect the dispersion of rGO.

On this basis, the dispersibility of the rGO used, which is particularly effective, differs from that obtained with other GRM, which generally require special compatibilization to improve their dispersion in the polymer matrix [4,13]. In addition to the type of GRM, the specific properties of the polymer matrix must also be taken into account. Indeed, it was reported that the exfoliation of graphite is promoted in solvents whose surface energy is equal to that of graphene [33]. As reported in a work on the preparation of nanocomposites based on PCL and GNP [34], for a polymer with a similar molecular weight to that used in this work, the surface tension of the molten PCL (51 mN/m) was found to be in the range of that of solvents such as dimethylformamide (DMF) and N-methyl-2-pyrrolidone (NMP), which have a high ability to exfoliate graphite.

It can be concluded that both the polymer matrix and the graphite used, which is characterized by a high surface area, as well as the preparation conditions turned out to promote the formation of homogeneous dispersed rGO-based systems.

To further investigate the dispersion of rGO and evaluate the effect of the nanoparticles on the dynamic network, rheological analyses were carried out on both PCL-G and PCL-OH_MDI networks. Storage modulus and complex viscosity for PCL-G (Appendix A) showed limited changes compared to the rheological behaviour of pristine PCL (Appendix A), with higher G’ and η* and no significant variation in the shape of the plots, which is consistent with the low concentration of rGO, below the percolation threshold. On the other hand, the inclusion of rGO within the dynamically crosslinked matrices caused dramatic changes in the rheological behavior. In fact, for both PCL-OH_MDI ratios, rGO induces a dramatic increase in both the storage modulus and viscosity plots (Figure 5), in particular at low frequencies, compared to the reference dynamically crosslinked networks, reaching remarkable values up to 10^5^ Pa in G’ and up to 10^6^ Pa∙s in η*, at 0.1 rad/s. In addition, G’ turns out to be almost independent of frequency in the whole range explored, which represents a close-to-ideal solid-like behavior, caused by very strong interactions within the network, promoted by the presence of rGO. Furthermore, modulus and viscosity values for nanostructured dynamic networks unexpectedly increase with increasing temperature, which is explained by a further evolution of the network towards higher strength. It is worth mentioning that this particular temperature evolution was not observed in the unfilled networks, suggesting rGO to produce additional crosslinking points, either by a specific interaction between the polymer chains and the surfaces or by confinement of macromolecules. While the clarification of the mechanism behind this peculiar rheological behavior would require further investigation, which is beyond the scope of this paper, it is worth underlighting that the inclusion of nanoparticles within covalent associative networks may be exploited to strongly modify the dynamics of network evolution, while preserving their reprocessability by compression molding, eventually enlarging the possible application of vitrimeric materials.

The thermal behavior of the composite samples was compared with that of PCL-OH and PCL (Appendix A and Table 4). Considering the data reported in Table 4, PCL-G showed an enhancement in T_c_ and *X_c_* compared to the neat polymer. The peculiar behavior of the above composite system might be ascribed to the effect of rGO on the polymer structuring. Indeed, as widely reported, the GRM can act as a nucleating agent for PCL [35], an effect that is particularly evident in the systems characterized by a high affinity for the graphitic surface. In the case of the PCL-OH_MDI_1:1_200_G and PCL_MDI_1:1.33_200_G, the crosslinking was found to have a slight effect on the polymer structuring and on T_c_, the decrease in crystallinity was not as severe as in the neat crosslinked samples described previously. Indeed, for these samples, the effect of rGO should also be considered, which could affect the thermal behavior of the samples and limit the decrease in T_c_ and *X_c_*.

### 3.3. Recycling and Self-Healing Tests

The nanocomposite sample PCL-OH_MDI_1:1.33_200_G was subjected to recycling and self-healing tests to investigate the effect of rGO on the processability of vitrimers. We focused on the above sample because it gave the most interesting results from a rheological point of view. In the recycling test, the sample was cut into small pieces and undergone high-temperature compression. As evident from the photo shown in Figure 6, it is possible to obtain a piece with controlled thickness and dimensions after the treatment. This demonstrates the process of the nanostructured dynamic network into simple shapes, as well as the possibility to recycle vitrimeric components at their end of life, in a circular approach [36]. Furthermore, the same sample also was found to have good adhesion ability when cut and put in contact at high temperatures. It is clear that a conventionally crosslinked materialdoes not have similar adhesion properties, nor it can be recycled by molding due to the low mobility of the network. Thus, our results indicate that the bond exchange between the different surfaces of the system can progress sufficiently and new bonds are formed between the surfaces. This suggests possible application in chemical welding of polymers [37], which is particularly relevant for polymer composites based on graphene related materials, as dispersed GRM may strongly reduce the possibility of conventional physical welding, either by restriction to the polymer chain diffusion or the limited adhesion of polymers onto GRM.

## 4. Conclusions

In this work, crosslinked materials characterized by dynamic covalent bonds based on PCL and rGO were developed. The preparation of these systems proves to be simple and easily scalable, since it is based on the use of a star-shaped hydroxyl-terminated PCL, whose polymerization can be carried out in bulk, and the well-established chemistry of isocyanates. Indeed, by simply changing the ratio between the -OH and -NCO functionalities and the process temperature, the degree of crosslinking as well as the thermal and rheological properties can be tuned. The composite materials turn out to be insoluble in solvents typical of the starting PCL, but are characterized by interesting recycling and self-healing properties. The presence of rGO also allows to strongly modify the dynamics of network evolution, while preserving their reprocessability and consequently their recyclability, eventually enlarging the field of application for vitrimeric materials.

## Figures and Tables

**Figure 1 nanomaterials-12-03744-f001:**
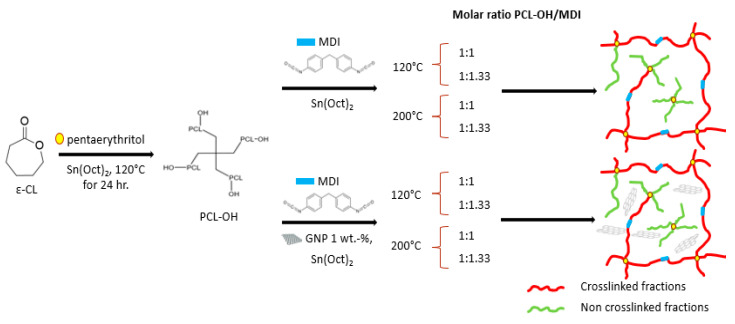
Scheme of the sample preparation.

**Figure 2 nanomaterials-12-03744-f002:**
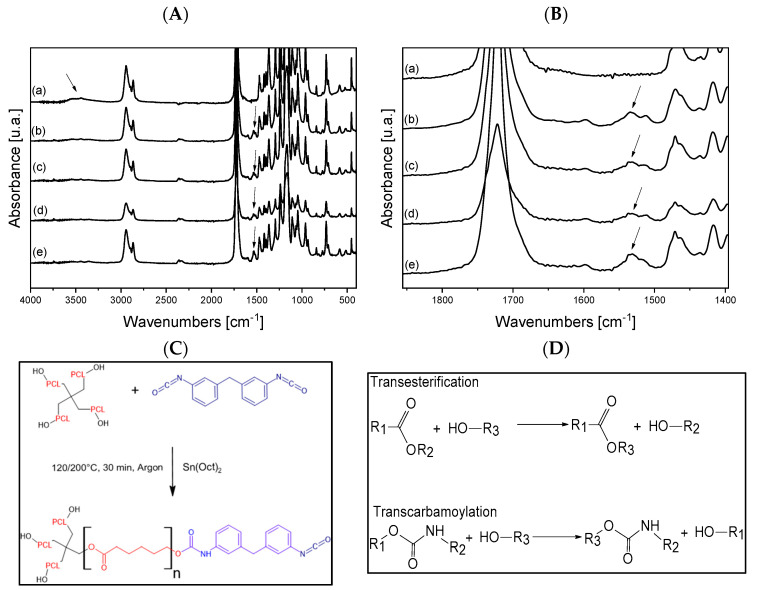
(**A**) FT-IR spectra of: (**a**) PCL-OH, (**b**) PCL-OH_MDI_1:1_120, (**c**) PCL-OH_MDI_1:1_200, (**d**) PCL-OH_MDI_1:1.33_120 and (**e**) PCL-OH_MDI_1:1.33_200. (**B**) Magnification of the FT-IR spectra in the secondary amide group region. (**C**) Scheme of the reaction between PCL-OH and MDI. (**D**) Scheme of the possible reactions within the network.

**Figure 3 nanomaterials-12-03744-f003:**
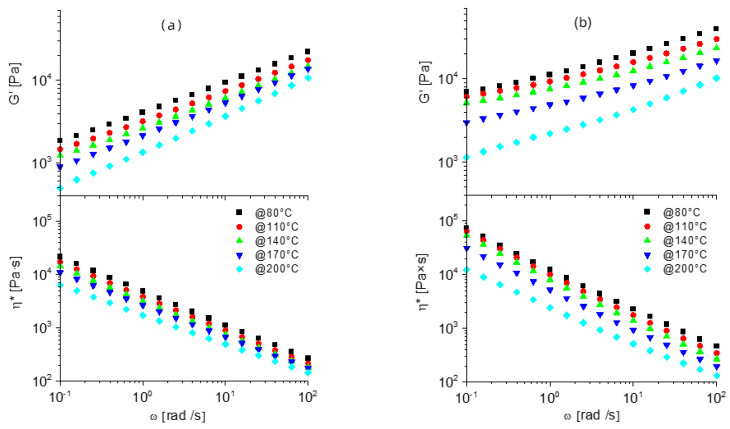
Rheological analyses plots, i.e., storage modulus (G’) and complex viscosity (η*) for (**a**) PCL-OH_MDI_1:1_120, (**b**) PCL-OH_MDI_1:1.33_120, (**c**) PCL-OH_MDI_1:1_200_and (**d**) PCL-OH_MDI_1:1.33_200.

**Figure 4 nanomaterials-12-03744-f004:**
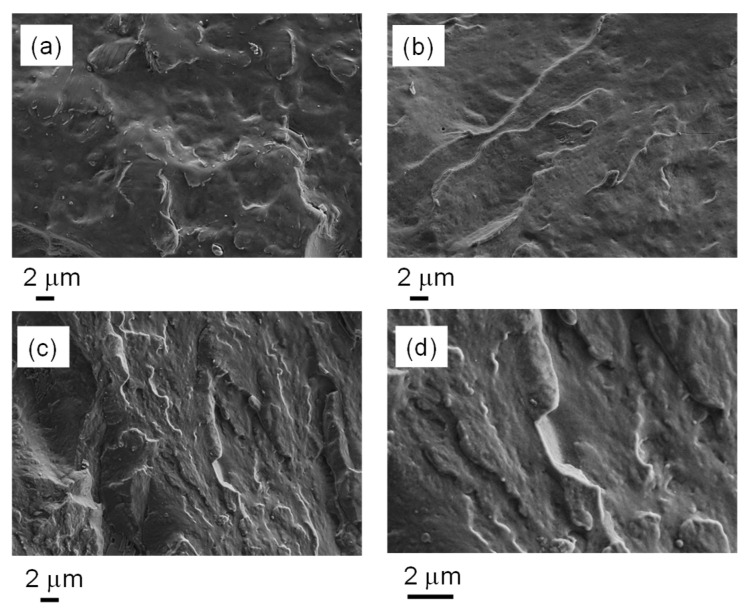
FE-SEM micrographs of: (**a**) PCL-G, (**b**) PCL-OH_MDI_1:1_200_G, (**c**) PCL-OH_MDI_1:1.33_200_G and (**d**) detail at high magnification of PCL-OH_MDI_1:1.33_200.

**Figure 5 nanomaterials-12-03744-f005:**
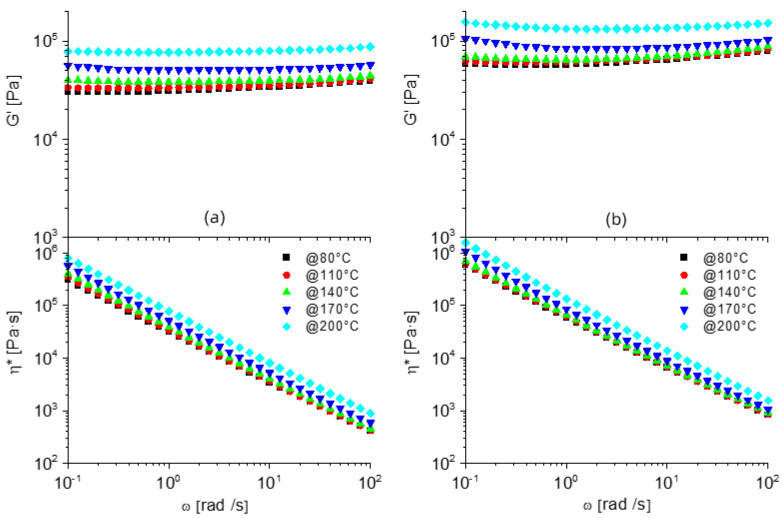
FE-SEM Rheological analyses plots, i.e., storage modulus (G’) and complex viscosity (η*) for (**a**) PCL-OH_MDI_1:1_200_G_and (**b**) PCL-OH_MDI_1:1.33_200_G_.

**Figure 6 nanomaterials-12-03744-f006:**
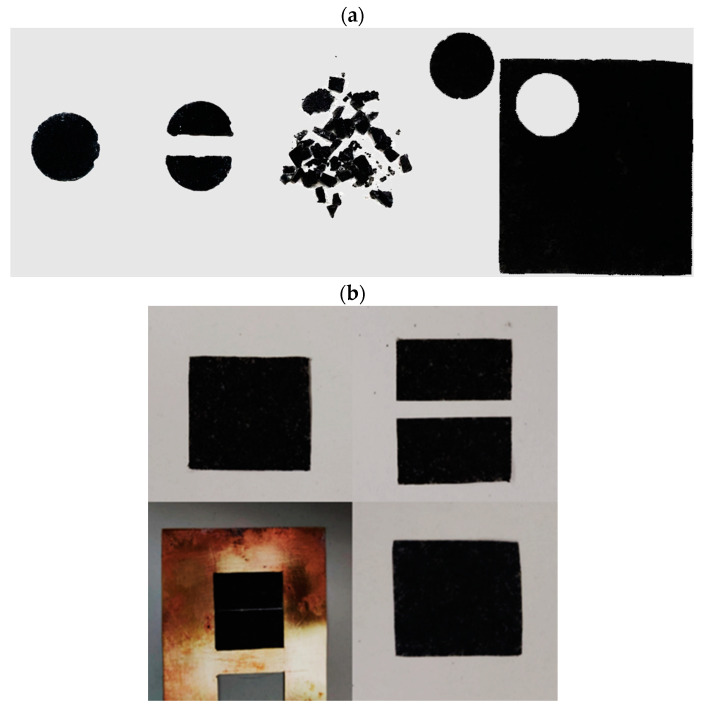
Properties of the sample PCL-OH_MDI_1:1.33_200_G: (**a**) recyclability (photos of the sample cut in two and small pieces and the resulting compressed film) and (**b**) self-adhesion (photos of the initial compressed film, the film divided into two pieces in the mold and the resulting compressed film).

**Table 1 nanomaterials-12-03744-t001:** Sample preparation conditions.

Sample Code	Molar Ratio PCL-OH:MDI	Molar Ratio OH:NCO	Temp. [°C]	rGO [wt.-%]
PCL-OH_MDI_1:1_120	1:1	2:1	120	-
PCL-OH_MDI_1:1.33_120	1:1.33	1.5:1	120	-
PCL-OH_MDI_1:1_200	1:1	2:1	200	-
PCL-OH_MDI_1:1.33_200	1:1.33	1.5:1	200	-
PCL-OH_MDI_1:1_120_G	1:1	2:1	120	1
PCL-OH_MDI_1:1.33_120_G	1:1.33	1.5:1	120	1
PCL-OH_MDI_1:1_200_G	1:1	2:1	200	1
PCL-OH_MDI_1:1.33_200_G	1:1.33	1.5:1	200	1

**Table 2 nanomaterials-12-03744-t002:** Gel fraction (*GF*) and swelling ratio (*SR*) of the neat crosslinked and nanocomposite samples.

Sample Code	*GF* [%]	*SR* [%]
PCL-OH_MDI_1:1_120	24	2600
PCL-OH_MDI_1:1.33_120	43	3300
PCL-OH_MDI_1:1_200	36	2800
PCL-OH_MDI_1:1.33_200	73	3200

**Table 3 nanomaterials-12-03744-t003:** Thermal properties of neat PCL and PCL-OH and of the prepared samples.

Sample Code	ΔH_c_ [J/g]	T_c_ [°C]	ΔH_m_ [J/g]	T_m_ [°C]	*X*_c_ [%]
PCL	−65	30	69	57	50
PCL-OH	−71	28	80	49	58
PCL-OH_MDI_1:1_120	−72	19	76	52	55
PCL-OH_MDI_1:1.33_120	−64	17	68	52	49
PCL-OH_MDI_1:1_200	−69	23	76	49	55
PCL-OH_MDI_1:1.33_200	−60	17	64	49	46

**Table 4 nanomaterials-12-03744-t004:** Thermal properties of neat PCL and of the prepared nanocomposites.

Sample Code	ΔH_c_ [J/g]	T_c_ [°C]	ΔH_m_ [J/g]	T_m_ [°C]	χ_c_ [%]
PCL	−65	30	69	57	50
PCL-OH	−71	28	80	49	58
PCL-G	−67	34	75	56	54
PCL-OH_MDI_1:1_200_G	−72	26	77	53	55
PCL-OH_MDI_1:1.33_200_G	−74	28	76	50	55

ΔH_c_ = crystallization enthalpy, T_c_ = crystallization temperature, ΔH_m_ = melting enthalpy, T_m_ = melting temperature, χ_c_ = crystallinity percentage.

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
