# Peer review of "On the Development of Nanocomposite Covalent Associative Networks Based on Polycaprolactone and Reduced Graphite Oxide"

_nanomaterials, 2022, doi:10.3390/nano12213744_

Round 1
Reviewer 1 Report
The author produced crosslinked materials by dynamic covalent bonds based on a star-shaped hydroxyl-terminated PCL and rGO. By simply changing the ratio between the -OH and -NCO functionalities and the process temperature, the degree of crosslinking as well as the thermal and rheological properties can be tuned. However, minor revisions are in need to make it more convinced.
- Why two molar ratios of PCL-OH:MDI (1:1 and 1:1.33) were chosen? What about 1.33:1?
- Why two temperatures (120 and 200 °C) were chosen? What about 100 °C?
- In page 10, the “e” label of Figure 4 mismatches with the corresponding figure caption. Please correct it.
- What is the product yield rate of PCL-OH_MDI_1:1.33_200_G under the lab-scale production that developed in this manuscript?
- Please compare the recycling and self-healing of PCL-OH_MDI_1:1.33_200_G with other recent publications to further stress the advantage of this facile production method.
Author Response
The author produced crosslinked materials by dynamic covalent bonds based on a star-shaped hydroxyl-terminated PCL and rGO. By simply changing the ratio between the -OH and -NCO functionalities and the process temperature, the degree of crosslinking as well as the thermal and rheological properties can be tuned. However, minor revisions are in need to make it more convinced.
- Why two molar ratios of PCL-OH:MDI (1:1 and 1:1.33) were chosen? What about 1.33:1?
The above molar ratios were used in order to prepare a material characterized by free hydroxyl functionalities to promote transesterification and transcarbamoylation reactions in the system. Indeed, as indicated in the Table 1, PCL-OH:MDI 1:1 and 1:1.33 correspond to OH:NCO ratios of 2:1 and 1.5:1, respectively. An excess hydroxyl function is known to be needed in this system to promote the exchange reaction but of course going for very large excess of hydroxyl would limit the crosslinking of the materials. In fact, based on the results obtained, PCL-OH:MDI 1:1 prepared at both temperatures exhibited rather low gel fractions and further increase in the OH:NCO ratios would result in even lower crosslinking, which is not of interest in this work, aiming at the preparation of covalent associative networks. In order to better assess this issue, the following sentence has been added in the text: “Four different samples were prepared at different temperatures (120 °C and 200 °C) and two molar ratios of PCL-OH:MDI, namely 1:1 and 1:1.33 (Table 1), i.e., under conditions which allow the production of a material with suitable concentration of -OH groups to promote bond exchange.”
- Why two temperatures (120 and 200 °C) were chosen? What about 100 °C?
120 °C was chosen to achieve a temperature far enough from PCL melting (50°C), also taking into account that in a recent work of ours (Li et al. Polycaprolactone adsorption and nucleation onto graphite nanoplates for highly flexible, thermally conductive, and thermomechanically stiff nano-papers. ACS Appl. Mater. Interfaces 2021, 13, 59206-5922015), we have demonstrated that PCL can perform some structuring near 100°C. In addition, we also selected a much higher temperature ( (200°C), to accelerate reactions and to better evidence the effect of this parameter on the material final features.
- In page 10, the “e” label of Figure 4 mismatches with the corresponding figure caption. Please correct it.
We thank the Reviewer for the accurate paper checking. The above mistake has been corrected.
- What is the product yield rate of PCL-OH_MDI_1:1.33_200_G under the lab-scale production that developed in this manuscript?
For the nanocomposite samples, which were insoluble and swollen in DCM, the data of the gel fraction, namely the yield of crosslinking, were not reported for the difficulty in performing the weighing, because rGO, even if finely dispersed in the polymer matrix, does not swell in the solvent like the polymer and this determines the brittleness of the material, making the weighing not effective.
- Please compare the recycling and self-healing of PCL-OH_MDI_1:1.33_200_G with other recent publications to further stress the advantage of this facile production method.
Recyclability and self-healing of vitrimers was widely reported in the literature and exchange mechanisms are quite well known. The aim of this paper was to demonstrate the recyclability and self-healing properties are retained in the presence of dispersed GRM, which was not reported before in the literature. Based on the reviewer comment, the text has been modified as follows to clarify the interest and advantages of the proposed materials:
“The nanocomposite sample PCL-OH_MDI_1:1.33_200_G was subjected to recycling and self-healing tests to investigate the effect of rGO on the processability of vitrimers. We focused on the above sample because it gave the most interesting results from a rheological point of view. In the recycling test, the sample was cut into small pieces and undergone to high temperature compression. As evident from the photo shown in Figure 6, it is possible to obtain a piece with controlled thickness and dimensions after the treatment. This demonstrates the process the nanostructured dynamic network into simple shapes, as well as the possibility to recycle vitrimeric components at their end of life, in a circular approach [36]. Furthermore, the same sample also was found to have good adhesion ability when cut and put in contact at high temperature. It is clear that a conventionally crosslinked material does not have similar adhesion properties, nor it can be recycled by molding due to the low mobility of the network. Thus, our results indicate that the bond exchange between the different surfaces of the system can progress sufficiently and new bonds are formed between the surfaces. This suggests possible application in chemical welding of polymers [37], which is particularly relevant for polymer composites based on graphene related materials, as dispersed GRM may strongly reduce the possibility of conventional physical welding, either by restriction to the polymer chain diffusion and the limited adhesion of polymers onto GRM.”
Reviewer 2 Report
In the manuscript entitled "On the development of nanocomposite covalent associative networks based on polycaprolactone and reduced graphite oxide" the development of nanocomposite systems based on reduced graphite oxide is presented. The crosslinking reaction was carried out to create systems capable of bond exchanges via transesterification and transcarbamoylation reactions. This work is very interesting and well-written. The manuscript is well-organized. However, several issues should be addressed before publication:
(1) In the literature review several works on the composites based on covalent networks, for example [Materials Today Physics (2022) 100851 (https://doi.org/10.1016/j.mtphys.2022.100851); Carbon (2022) 191, P. 55] can be also mentioned since it shows the actuality of this subject in different areas;
(2) Authors wrote (line 262): "Moreover, one new signal appear at 1535 cm-1 in these samples (Figure 2B), which can be attributed to the combinations of C-N stretching and N-H bending of the secondary amide groups [31]." This should be additionally described.
(3) line "It is worth underling that, as described in the literature for other materials [23,24]..." Again, especially which materials are considered in [23,24], and how they close to the considered systems? Further, the Authors compare the results with works [32-34].
(4) Additional discussion with the literature is required.
Author Response
In the manuscript entitled "On the development of nanocomposite covalent associative networks based on polycaprolactone and reduced graphite oxide" the development of nanocomposite systems based on reduced graphite oxide is presented. The crosslinking reaction was carried out to create systems capable of bond exchanges via transesterification and transcarbamoylation reactions. This work is very interesting and well-written. The manuscript is well-organized. However, several issues should be addressed before publication:
-In the literature review several works on the composites based on covalent networks, for example [Materials Today Physics (2022) 100851 (https://doi.org/10.1016/j.mtphys.2022.100851); Carbon (2022) 191, P. 55] can be also mentioned since it shows the actuality of this subject in different areas;
We thank the reviewer for the suggestion, but after careful reading of the above references we did not see a clear relation of the papers to our work. Taking into account of the editorial requirement to limit the citations to the most important and relevant ones, we prefer not to include in the text the two proposed by the reviewer.
- Authors wrote (line 262): "Moreover, one new signal appear at 1535 cm-1 in these samples (Figure 2B), which can be attributed to the combinations of C-N stretching and N-H bending of the secondary amide groups [31]." This should be additionally described.
We thank the Reviewer for the valuable suggestion, the above sentence has been modified as follows: “Moreover, one new signal appear at 1535 cm-1 in these samples (Figure 2B), which can be attributed to N-H bending of the secondary amide groups.”
- line "It is worth underling that, as described in the literature for other materials [23,24]..." Again, especially which materials are considered in [23,24], and how they close to the considered systems? Further, the Authors compare the results with works [32-34].
The material considered in references 23 and 24 has been specified in text as follows: “It is worth underling that, as described in the literature for other polyesters, such as polylactic acid [23,24], i.e., polymers with a similar chemical structure to PCL, the presence of the catalyst can activate transesterification and transcarbamoylation reactions involving free hydroxyl functionalities of the system (Figure 2D), thus forming a covalent associative network.”
In addition, the comparison with references 32, 33 and 34 has been better clarified as follows:
“The neat PCL showed a typical behavior of a semi-crystalline polymer, with crystallinity (Xc), melting (Tm) and crystallization temperatures (Tc) in agreement with the data reported in the literature for a polymer with similar characteristics [32].”
“In addition to the type of GRM, the specific properties of the polymer matrix must also be taken into account. Indeed, it was reported that the exfoliation of graphite is promoted in solvents whose surface energy is equal to that of graphene [33]. As reported in a work on the preparation of nanocomposites based on PCL and GNP [34], for a polymer with a similar molecular weight to that used in this work, the surface tension of the molten PCL (51 mN/m) was found to be in the range of that of solvents such as dimethylformamide (DMF) and N-methyl-2-pyrrolidone (NMP), which have a high ability to exfoliate graphite.”
- Additional discussion with the literature is required
Performance of nanostructured vitrimers especially in terms of recyclability and self-healing was further discussed within the frame of recent scientific literature.
Indeed, while recycling and self-healing of vitrimers was widely reported in the literature and exchange mechanisms are quite well known, in this paper we demonstrate the recyclability and self-healing properties are retained in the presence of dispersed GRM, which was not reported before in the literature. Based on the reviewer comment, the text was modified as follows to clarify this point and refer to the recent literature:
“The nanocomposite sample PCL-OH_MDI_1:1.33_200_G was subjected to recycling and self-healing tests to investigate the effect of rGO on the processability of vitrimers. We focused on the above sample because it gave the most interesting results from a rheological point of view. In the recycling test, the sample was cut into small pieces and undergone to high temperature compression. As evident from the photo shown in Figure 6, it is possible to obtain a piece with controlled thickness and dimen-sions after the treatment. This demonstrates the process the nanostructured dynamic network into simple shapes, as well as the possibility to recycle vitrimeric components at their end of life, in a circular approach [36]. Furthermore, the same sample also was found to have good adhesion ability when cut and put in contact at high temperature. It is clear that a conventionally crosslinked material does not have similar adhesion properties, nor it can be recycled by molding due to the low mobility of the network. Thus, our results indicate that the bond exchange between the different surfaces of the system can progress sufficiently and new bonds are formed between the surfaces. This suggests possible application in chemical welding of polymers [37], which is particularly relevant for polymer composites based on graphene related materials, as dispersed GRM may strongly reduce the possibility of conventional physical welding, either by restriction to the polymer chain diffusion and the limited adhesion of polymers onto GRM.”
Reviewer 3 Report
The present work report the crosslinked materials of composite of PCL and rGO, which showed tunable cross-linking and thermal properties by changing the ratio between the -OH and -NCO and the process temperature. The result is quite interesting and it can be published with minor reversion. The author can clarify the following items before publication:
(1) Figure 2d, please use the draw the reaction with the chemical structures used in the present work, not a general chemical reaction scheme
(2) Please elaborate more details of the recycling and self-healing tests
(3) For Fig6a and b, are they schematic diagrams or experimental results photos? Please described clearly in the fig caption. If they are the photos, please use better resolution ones
Author Response
The present work report the crosslinked materials of composite of PCL and rGO, which showed tunable cross-linking and thermal properties by changing the ratio between the -OH and -NCO and the process temperature. The result is quite interesting and it can be published with minor reversion. The author can clarify the following items before publication:
- Figure 2d, please use the draw the reaction with the chemical structures used in the present work, not a general chemical reaction scheme
As suggested a detailed draw of the reaction has been reported in the Supporting Information (Figure S2).
- Please elaborate more details of the recycling and self-healing tests
The part dealing with the recycling and self-healing has been rewrote as follows:
The nanocomposite sample PCL-OH_MDI_1:1.33_200_G was subjected to recycling and self-healing tests to investigate the effect of rGO on the processability of vitrimers. We focused on the above sample because it gave the most interesting results from a rheological point of view. In the recycling test, the sample was cut into small pieces and undergone to high temperature compression. As evident from the photo shown in Figure 6, it is possible to obtain a piece with controlled thickness and dimensions after the treatment. This demonstrates the process the nanostructured dynamic network into simple shapes, as well as the possibility to recycle vitrimeric components at their end of life, in a circular approach [36]. Furthermore, the same sample also was found to have good adhesion ability when cut and put in contact at high temperature. It is clear that a conventionally crosslinked material does not have similar adhesion properties, nor it can be recycled by molding due to the low mobility of the network. Thus, our results indicate that the bond exchange between the different surfaces of the system can progress sufficiently and new bonds are formed between the surfaces. This suggests possible application in chemical welding of polymers [37], which is particularly relevant for polymer composites based on graphene related materials, as dispersed GRM may strongly reduce the possibility of conventional physical welding, either by restriction to the polymer chain diffusion and the limited adhesion of polymers onto GRM
- For Fig6a and b, are they schematic diagrams or experimental results photos? Please described clearly in the fig caption. If they are the photos, please use better resolution ones
The above Figure caption has been changed as follows: Properties of the sample PCL-OH_MDI_1:1.33_200_G: (a) recyclability (photos of the sample cut in two and in small pieces and the resulting compressed film) and (b) self-adhesion (photos of the initial compressed film, the film divided in two pieces in the mold and the resulting compressed film). Moreover, the resolution of the photos has been improved.